# Tumor reactive γδ T cells contribute to a complete response to PD-1 blockade in a Merkel cell carcinoma patient

Scott C. Lien [1,2], Dalam Ly [1], S. Y. Cindy Yang[1], Ben X. Wang[1], Derek L. Clouthier[1], Michael St. Paul [1], Ramy Gadalla[1], Babak Noamani[1], Carlos R. Garcia-Batres [1], Sarah Boross-Harmer[3], Philippe L. Bedard [3], Trevor J. Pugh [1,4,5], Anna Spreafico [3], Naoto Hirano[1,2], Albiruni R. A. Razak[3] & Pamela S. Ohashi [1,2] ✉

Immunotherapies targeting PD-1/PD-L1 are now widely used in the clinic to treat a variety of malignancies. While most of the research on T cell exhaustion and PD-1 blockade has been focused on conventional αβ T cells, the contribution of innate-like T cells such as γδ T cells to anti-PD-1/PD-L1 mediated therapy is limited. Here we show that tumor reactive γδ T cells respond to PD-1 blockade in a Merkel cell carcinoma (MCC) patient experiencing a complete response to therapy. We find clonally expanded γδ T cells in the blood and tumor after pembrolizumab treatment, and this Vγ2Vδ1 clonotype recognizes Merkel cancer cells in a TCR-dependent manner. Notably, the intra-tumoral γδ T cells in the MCC patient are characterized by higher expression of PD-1 and TIGIT, relative to conventional CD4 and CD8 T cells. Our results demonstrate that innate-like T cells could also contribute to an anti-tumor response after PD-1 blockade.

γδ T cells are innate-like T cells that are activated by a wide variety of molecules through TCR-dependent and independent mechanisms. These ligands include phosphoantigens presented on butyrophilins[1,2], MHC class I-like family members MR1[3] and CD1c[4], as well as stress-induced molecules such as MICA[5]. However, γδ T cells have also been shown to recognize the MART-1 tumor antigen in an MHC class I-restricted and TCR-dependent manner[6].

Landmark studies have shown that γδ T cells are important for tumor-surveillance in animal models of skin cancer and lymphoma[7,8]. Moreover, in a comprehensive pan-cancer analysis spanning 25 distinct cancer types, tumor-infiltrating γδ T cells were the immune cell subset exhibiting the most pronounced association with a favorable prognosis[9]. This observation notably surpasses the prognostic significance of CD8 and CD4 T cell infiltration, which has been well established by immunohistochemistry analysis of various cancer types[10–12]. Despite PD-1 blockade being utilized as a cornerstone therapeutic strategy to target conventional αβ T cells, the effect of PD-1 blockade on γδ T cells remains overlooked.

Like αβ T cells, γδ T cells can also express PD-1 upon TCR stimulation with phosphoantigens and ectopic expression of PD-L1 dampens effector functions of PD-1⁺ T cells[13]. Additionally, in patients with multiple myeloma, Vγ9Vδ2 T cells found in the bone marrow are PD-1⁺ and have reduced proliferative capacity in response to phosphoantigen stimulation as compared to Vγ9Vδ2 T cells found in peripheral blood, suggesting dysfunction of Vγ9Vδ2 T cells due to chronic TCR stimulation[14]. Furthermore, in models of adoptive cell transfer with Vγ9Vδ2 T cells, anti-PD-1 treatment can enhance tumor control by either antibody-dependent cellular cytotoxicity[15] or recognition of phosphoantigens[16].

[1]Princess Margaret Cancer Centre, University Health Network, Toronto, ON, Canada. [2]Department of Immunology, University of Toronto, Toronto, ON, Canada. [3]Division of Medical Oncology and Haematology, Princess Margaret Cancer Centre, University Health Network, University of Toronto, Toronto, ON, Canada. [4]Department of Medical Biophysics, University of Toronto, Toronto, ON, Canada. [5]Ontario Institute for Cancer Research, Toronto, ON, Canada. ✉e-mail: pam.ohashi@uhnresearch.ca

Most of the research on PD-1 and γδ T cells have been focused on the phosphoantigen-sensing Vγ9Vδ2 subset, and there is less evidence supporting the contribution of non-Vδ2 T cells to PD-1 blockade. Recent studies in patients treated with checkpoint blockade have demonstrated that increased signatures of non-Vδ2 T cells in kidney cancer patients were associated with response to treatment with atezolizumab (anti-PD-L1)[17] and that Vδ1/Vδ3 T cells are important effectors in MHC class I-deficient tumors[18].

To further examine whether γδ T cells are direct mediators of the therapeutic response to PD-1 blockade, we evaluate the γδ T cell responses in six Merkel cell carcinoma (MCC) patients treated with pembrolizumab. Here we show clonal expansion of tumor reactive γδ T cells in a MCC patient with a complete response after pembrolizumab treatment, thus supporting a significant role for γδ T cells in the response to PD-1 blockade.

## Results

### Expansion of γδ T cells in the tumor and blood after pembrolizumab treatment

To investigate the dynamic changes of the anti-tumor response during PD-1 blockade, we obtained tumor biopsies at baseline and on treatment from six MCC patients. One patient (E-013) had a complete response, two patients (E-020 and E-022) had partial responses, and three patients (E-033, E-034, and E-035) had progressive disease (see Supplementary Table 1 for patient details). We designed flow cytometry panels to characterize different T cell subsets and their phenotype (see Supplementary Fig. 1A for flow gating strategy). Surprisingly, we found almost 10-fold expansion (3.68 to 35.8%) in the frequency of γδ T cells in patient E-013 who experienced a complete response to therapy (Fig. 1A, B). Notably, the intra-tumoral γδ T cells in patient E-013 had higher expression of PD-1 and TIGIT relative to their conventional CD4 and CD8 T cells (Fig. 1C). This is in contrast to intra-tumoral lymphocytes from a patient with partial response, E-022, and patients with progressive disease (E-033 and E-035), where CD4 and CD8 T cells from patients E-022, E-033 and E-035 show a more typical profile with their CD4 and CD8 αβ T cells having higher expression of PD-1 and TIGIT. Comparatively, γδ T cells from patients that did not expand after treatment, did not have high expression of PD-1 or TIGIT, supporting the possibility that the expanded γδ cells in patient E-013 have received a γδ T cell-specific stimuli and were involved in the anti-tumor response.

We also evaluated the peripheral blood from these patients at baseline and several time points after treatment. Analysis of the peripheral blood from patient E-013 also showed a four-fold expansion of γδ T cells (1.4 to 7.28%) that peaked at week 12 and then subsequently declined (Fig. 2A). This expansion of γδ T cell frequency in blood and tumor (on-treatment tumor biopsy timepoint depicted with black arrow) interestingly coincided with a subsequent decrease in tumor burden. While Vγ9Vδ2 T cells are typically the predominant γδ T cell subset in peripheral blood[19], there was a wide range of Vδ2 cells (8.98–91.1%) amongst the MCC patients (Fig. 2B, C). We also observed that patients E-013 and E-035 had an increase in Vδ1 cells after pembrolizumab treatment with a corresponding decrease in Vδ2 cells (Fig. 2C). Both patients E-013 and E-035 had an increase in Ki-67+ γδ T cells at week 6 of anti-PD-1 treatment, indicating that the change in frequency of γδ T cell subsets was likely due to proliferation (Fig. 2D, E). Furthermore, this proliferation was primarily attributed to the Vδ1 subset. When we looked at PD-1 expression in peripheral blood, γδ T cells from patient E-013 had higher expression of PD-1 compared to their CD8 or CD4 T cells (Fig. 2F). Notably, the Vδ1 subset had increased expression of TIM-3 upon pembrolizumab treatment (Fig. 2G). In addition, γδ T cells from patient E-013 and E-035 had increased TIGIT expression, which coincided with a decrease in CD27 and CD28 expression (Supplementary Fig. 1B), suggestive of a more effector-like phenotype[20]. Together, these results show that in some

MCC patients, γδ T cells exhibit an activated phenotype and PD-1 blockade can result in proliferation and expansion of γδ T cell subsets in both the blood and tumor.

### Clonally expanded peripheral blood γδ T cells are detected in the tumor

Next, we wanted to examine whether anti-PD-1 treatment led to clonal expansion of γδ T cells, which would support the possibility that γδ T cells were specifically recognizing MCC. To address this question, we sorted γδ T cells from peripheral blood at baseline and week 6 and sequenced the TCRγ (TRG) chains to explore the diversity and clonal dynamics upon pembrolizumab treatment. Using D50 index as a measurement of diversity (minimum number of clonotypes that comprise 50% of the total repertoire), patient E-013 had a decrease in diversity (from 10 to 1 clonotypes), while E-035 had an increase in diversity (from 61 to 107 clonotypes) upon PD-1 blockade (Fig. 3A). When looking at the frequency of the top 10 clonotypes by CDR3 amino acid sequence, we observed that in patient E-013, one clonotype increased from 10.8 to 54.6% (Fig. 3B), while in patient E-035, the frequencies of the top 10 clonotypes shrank to comprise only 5.6% of the total repertoire (Fig. 3B). On the other hand, we observed patients E-020, E-022, and E-034 had minimal changes to their TRG repertoire (Fig. 3C), highlighting the significance of the clonal expansion observed from patient E-013.

We next assessed whether the expanded clonotypes in the E-013 blood could also be found within the tumor. We purified γδ T cells from week 6 PBMCs and compared the CDR3 sequences with genomic DNA extracted from FFPE tissue of a surgically resected lesion after the patient was taken off pembrolizumab after 34 cycles due to intercurrent illness. Because αβ T cells could potentially have an in-frame TRG rearrangement[21] and it is difficult to isolate γδ T cells from FFPE tissue, we utilized TCRδ (TRD) chain sequencing to obtain a more accurate comparison of the TCR repertoire between γδ T cells in peripheral blood and FFPE tissue. When observing the top 10 clonotypes, we found that the TRD repertoire is highly comparable between blood and tumor, with the top clonotype representing 76.3% in week 6 blood and 43.9% in the tumor (Fig. 3D). Taken together, these results suggest that a specific γδ T cell clonotype expanded and was found in both the blood and the tumor from a MCC patient who had a complete response to PD-1 blockade.

### Clonal expansion of cytotoxic γδ T cells determined by single cell analysis

To further examine the γδ TCR repertoire and phenotype, we leveraged single cell RNA and TCR sequencing to track specific clonotypes along with their associated gene expression profile. We sorted γδ T cells from week 6 peripheral blood of patients E-013 and E-035 and obtained 3,646 cells spread across eight clusters as determined by scRNAseq analysis (Fig. 4A). From the TCR sequencing, we identified clonally expanded γδ T cells (Fig. 4B) that were predominantly found in clusters 0, 1 and 2 (Fig. 4C). This particular clonotype was a Vγ2Vδ1 heterodimer that was identical to the top TRG and TRD CDR3 amino acid sequences found in our bulk TCR sequencing data (Fig. 3B, D), and we designated this clonotype as γδTCR3-4. (Fig. 4B). The γδTCR3-4 clonotype totaled 1,913 cells, while clonotypes from E-035 never surpassed 20 cells. When we looked at the most differentially expressed genes between each cluster, we found that cluster 0 were TRGV2+ cells that expressed ZFP36 (Fig. 4D), which may indicate recent TCR stimulation[22]. Cluster 1 cells have high expression of genes associated with MHC class II antigen presentation such as CD74 invariant chain, HLA-DRA and HLR-DRB1, which can be markers of T cell activation[23], while cytotoxic cells expressing GZMB, GZMH and GNLY are found in cluster 2 (Fig. 4D). Notably, cluster 5 was the most distinct cluster and comprised of TRGV9 and TRDV2 cells typically found in peripheral blood (Fig. 4E). Surprisingly, these Vγ9Vδ2 cells were a minor fraction

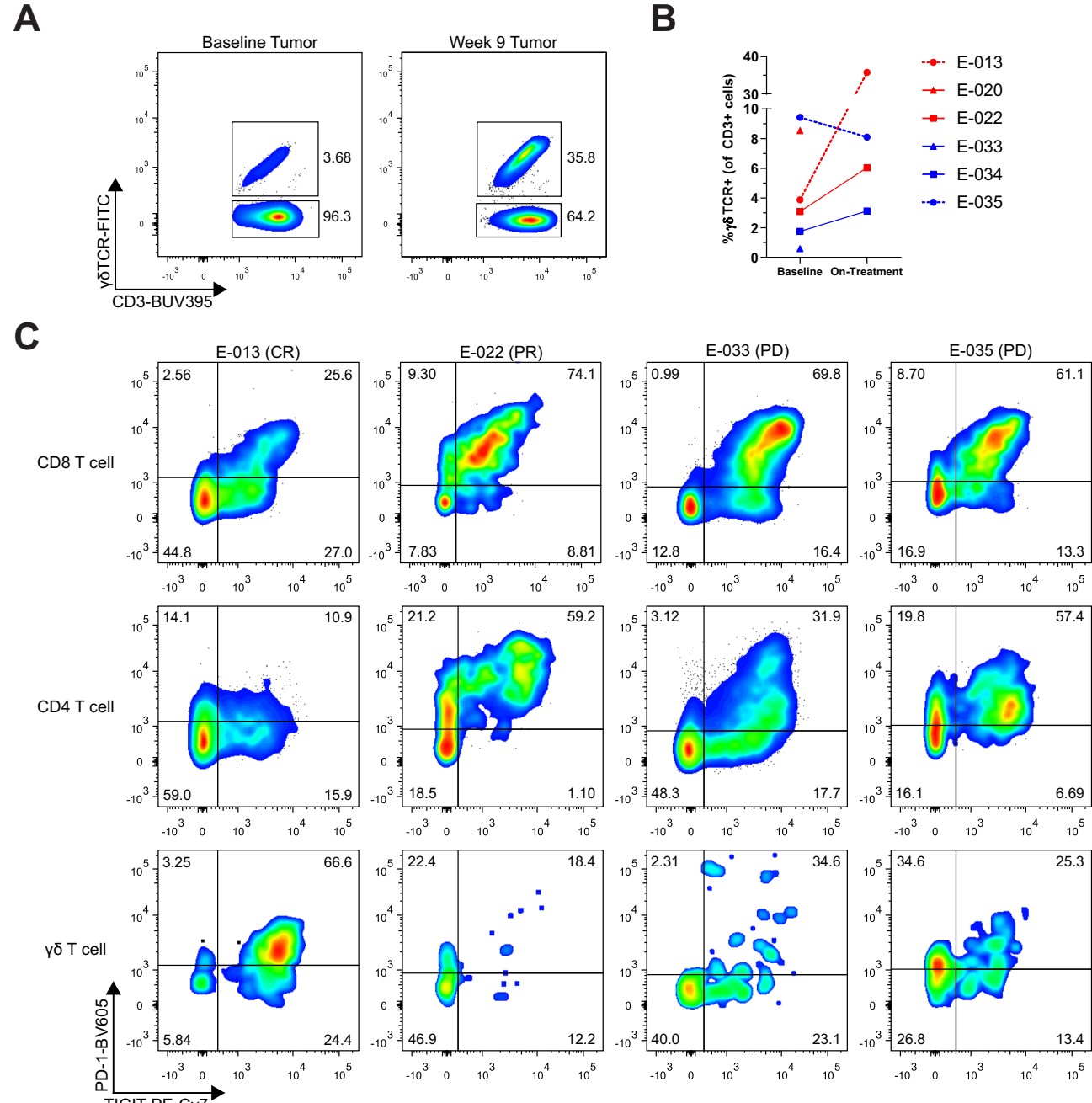

**Fig. 1 | γδ T cells were activated and expanded in the tumor upon pembrolizumab treatment in a complete responder MCC patient.** Single cell suspensions of tumor core biopsies from MCC patients were evaluated by flow cytometry. **A** Flow cytometry analysis of γδ T cells in baseline and week 9 tumor biopsies from the complete responder patient E-013 is shown. **B** Quantification of γδ T cell frequency as measured by flow cytometry in tumor biopsies from six MCC patients. Red denotes partial response (PR)/complete response (CR), blue represents progressive disease (PD). Each individual patient is marked by different symbols as shown. Data for on-treatment biopsy samples from patient E-020 and E-033 were unavailable due to insufficient tumor material. Source data provided as a source data file. **C** Lymphocytes from baseline biopsies of four MCC patients were evaluated for PD-1 and TIGIT expression and gated on CD8, CD4 and γδ T cells.

of cells and when we compared them with γδTCR3-4 cells, the most differentially expressed genes were maturation markers *IL7R* and *SELL* on Vγ9Vδ2 cells, representing a more naïve or central memory phenotype. On the other hand, γδTCR3-4 cells had higher expression of the effector molecules *IFNG* and *PRF1*, as well as the exhaustion molecule *TIGIT* (Fig. 4F). To further explore and potentially gain functional insights into the γδTCR3-4 clonotype, we subsetted out these cells and used *Slingshot* to perform trajectory inference analysis. We found the γδTCR3-4 clonotype branched into two distinct lineages. Across pseudotime, there was higher expression of *GZMK* in Lineage 2 with a concurrent increase in *GZMB* in both lineages (Fig. 4G, H). TCR

activation in *GZMK*+ γδ T cells was recently shown to initiate GZMK release and subsequently gain GZMB expression[24]. This indicates a proportion of the γδTCR3-4 cells were stimulated with antigen, gained cytotoxic potential and may be directly involved in clearing the tumor. In summary, these findings highlight that γδTCR3-4 cells from patient E-013 were clonally expanded and have cytotoxic potential.

**Expanded γδ T cell clonotype recognizes Merkel cell cancer**
To determine if γδTCR3-4 could recognize Merkel cancer cells, we cloned and expressed γδTCR3-4 into Jurkat76 cells that lacked the endogenous TCR and co-cultured them with various Merkel and

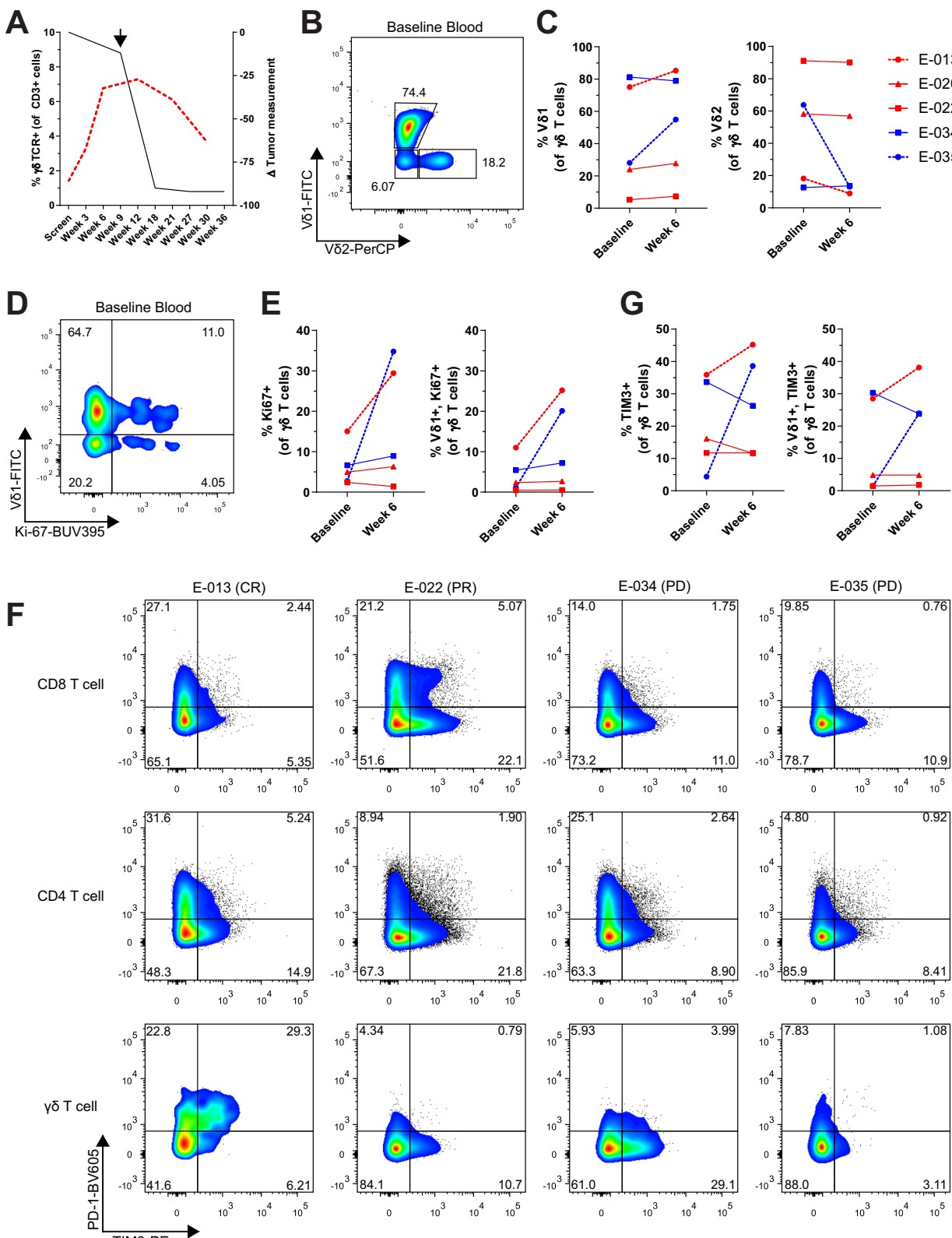

**Fig. 2 | Vδ1 T cells were activated and proliferate in peripheral blood after anti-PD-1 treatment in a complete responder MCC patient. A** Expansion of γδ T cells measured by flow cytometry is compared with tumor regression over time. The red dotted line represents γδ T cell frequency measured in peripheral blood from patient E-013. Tumor measurements from E-013 by CT scan are indicated by the black line. Arrow denotes when the on-treatment tumor biopsy was taken. **B** Flow cytometry analysis of Vδ1 and Vδ2 subsets gated on γδ T cells from patient E-013 baseline blood. **C** Frequency of Vδ1 cells (left) and Vδ2 cells (right) in peripheral blood at baseline and week 6 from five MCC patients as shown in the legend. **D** Proliferation gated on γδ T cells from baseline blood of E-013 was evaluated by flow cytometry. **E** Summary of Ki-67⁺ γδ T cells (left) and Ki-67⁺ Vδ1 cells (right) for five of the MCC patients as shown in the legend of (**C**). **F** Flow cytometry plots of PD-1 and TIM-3 expression gated on CD8, CD4 and γδ T cells from baseline blood of four MCC patients. **G** Summary of TIM-3⁺ γδ T cells and TIM-3⁺ Vδ1 cells in blood of five MCC patients. Source data for (**C**, **E** and **G**) provided as a source data file.

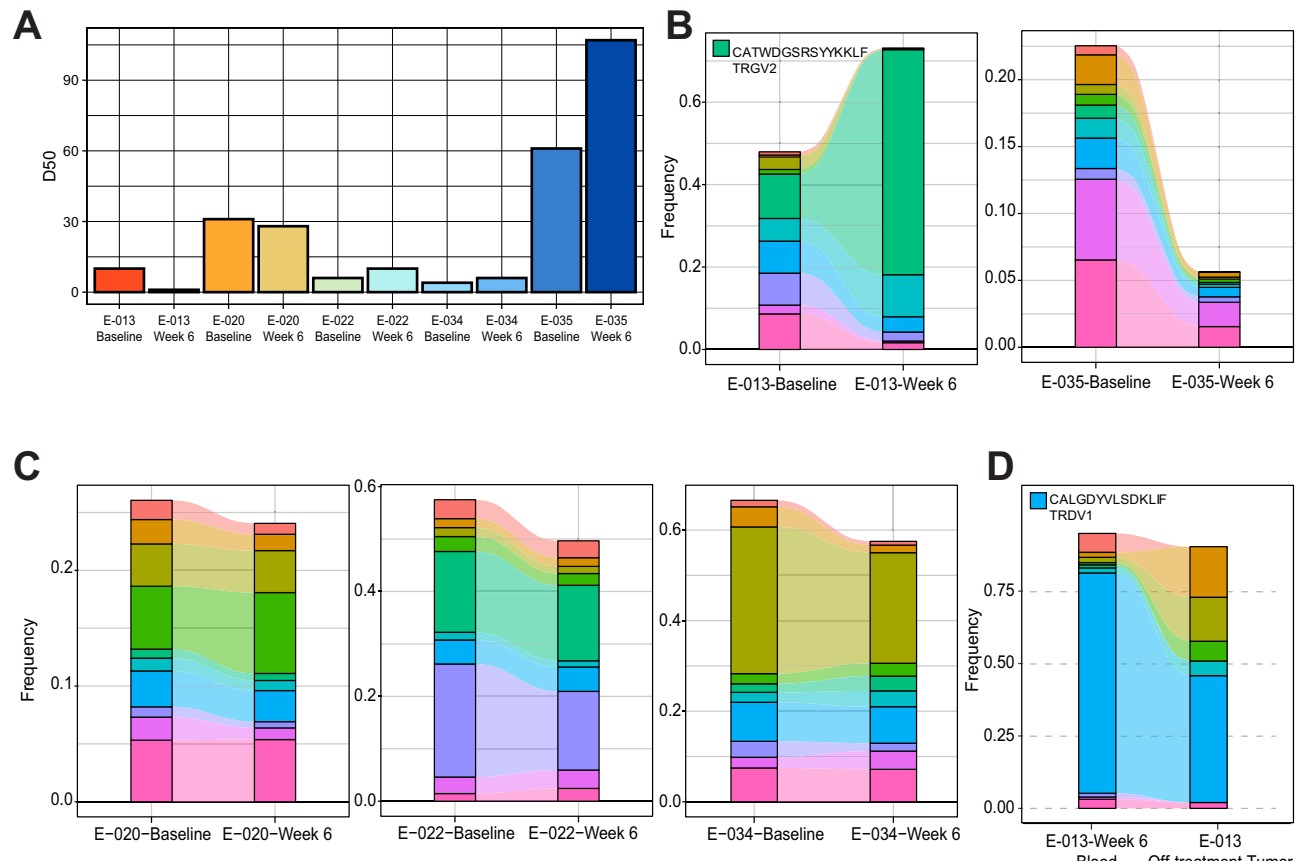

**Fig. 3 | Clonotypic response of γδ T cells in a MCC patient upon pembrolizumab treatment.** γδ T cells were sorted from the peripheral blood at baseline and week 6 after pembrolizumab treatment and TCRγ chains were sequenced. **A** Diversity 50 (D50) index on *TRG* CDR3 clonotypes from five MCC patients are shown. D50 index is the minimum number of unique clonotypes to occupy 50% of the total repertoire. **B** Changes in the frequency of top 10 clonotypes by CDR3 a.a. sequence for the complete responder patient E-013 (left) and progressive disease patient E-035 (right). The TCRγ CDR3 sequence CATWDGSRSYYKKLF is shown for the top

expanded clonotype from patient E-013. **C** Frequency of top 10 clonotypes for the partial responders E-020 (left), E-022 (middle), and progressive disease patient E-034 (right). **D** Comparison of the top 10 TCRδ CDR3 sequences of γδ T cells sorted from week 6 blood with TCRδ sequenced from an FFPE tumor sample taken when a new lesion arose in patient E-013 after pembrolizumab treatment was stopped due to intercurrent illness. The TCRδ CDR3 sequence CALGDYVLSDKLIF is shown for the most abundant clonotype.

ovarian cancer cell lines. As controls, we used the next most frequent clonotype to generate mis-paired γδTCR; either the γ chain (γδTCR3-5) or the δ chain (γδTCR1-4). All three γδ TCRs were stably expressed and maintained on Jurkat76 cells (Supplementary Fig. 2). Using CD69 upregulation as a readout of TCR reactivity, we found that while all γδTCR Jurkat76 cells responded to anti-CD3 stimulation, only γδTCR3-4 was able to recognize the Merkel cancer cell lines MCC14/2, MCC26 and MS-1, but not the ovarian cell lines OVCAR3 and CaOV3 (Fig. 5A, B). To further examine the specificity of the γδTCR3-4, we performed co-cultures with different ratios of Jurkat cells and tumor cells. When we co-cultured with all the Merkel cancer cell lines, only γδTCR3-4 but not the mis-paired γδ TCRs, nor the non-transduced Jurkat76 parental line had a dose-dependent increase in CD69 expression (Fig. 5C). These results demonstrate that γδTCR3-4, which expanded in patient E-013 during pembrolizumab treatment, recognized MCC.

### γδ T cell reactivity and specificity
To determine whether the γδTCR3-4 could recognize healthy tissues, we co-cultured γδTCR3-4 Jurkat76 cells with non-cancerous cells such as PBMCs and human umbilical vein endothelial cells (HUVEC) and found no detectable reactivity (Fig. 5D). Furthermore, myeloma and leukemic cell lines H929, THP1 and K562 did not induce CD69 expression in the γδTCR3-4 Jurkat76 cells (Fig. 5A). Unexpectedly, co-culture with melanoma cells

A375, 624mel and 888mel induced CD69 expression (Fig. 5A, D), suggesting that the stimulating ligand is shared between some cancers. Although both melanoma and MCC fall under the umbrella of skin cancers, these cancer types exhibit distinct origins, where melanocytes arise from the neural crest, while Merkel cells are believed to originate from the epidermal ectoderm[25].

To further investigate the specificity of the MCC reactive γδTCR3-4, we utilized CRISPR-Cas9 to knockout known γδ T cell ligands such as *B2M* (which is required for HLA class I, CD1 isoforms and MR1), *EPCR*, *EphA2*, *MICA/B*, *ANXA2* and *BTN3A1*. However, all knockout cell lines still demonstrated γδTCR3-4 reactivity comparable to scrambled gRNA control (Supplementary Fig. 3A–G). It was recently demonstrated that γδ T cells can recognize HLA-DR in CMV-infected cells[26], but the MCC26 cell line does not express HLA-DR, DP, or DQ (Supplementary Fig. 3H). Together these results demonstrate that the putative ligand recognized by γδTCR3-4 is potentially uncharacterized.

In order to examine whether γδ T cells are expanded after PD-1 blockade in other cancer types, we evaluated samples from the INSPIRE clinical trial which included squamous cell carcinoma of the head and neck, triple-negative breast cancer (TNBC), high-grade serous ovarian cancer, melanoma, and a cohort of mixed solid tumors[27]. Amongst a total of 30 paired baseline and on-treatment biopsies, flow cytometry analysis showed that an additional TNBC patient had approximately a six-fold increase

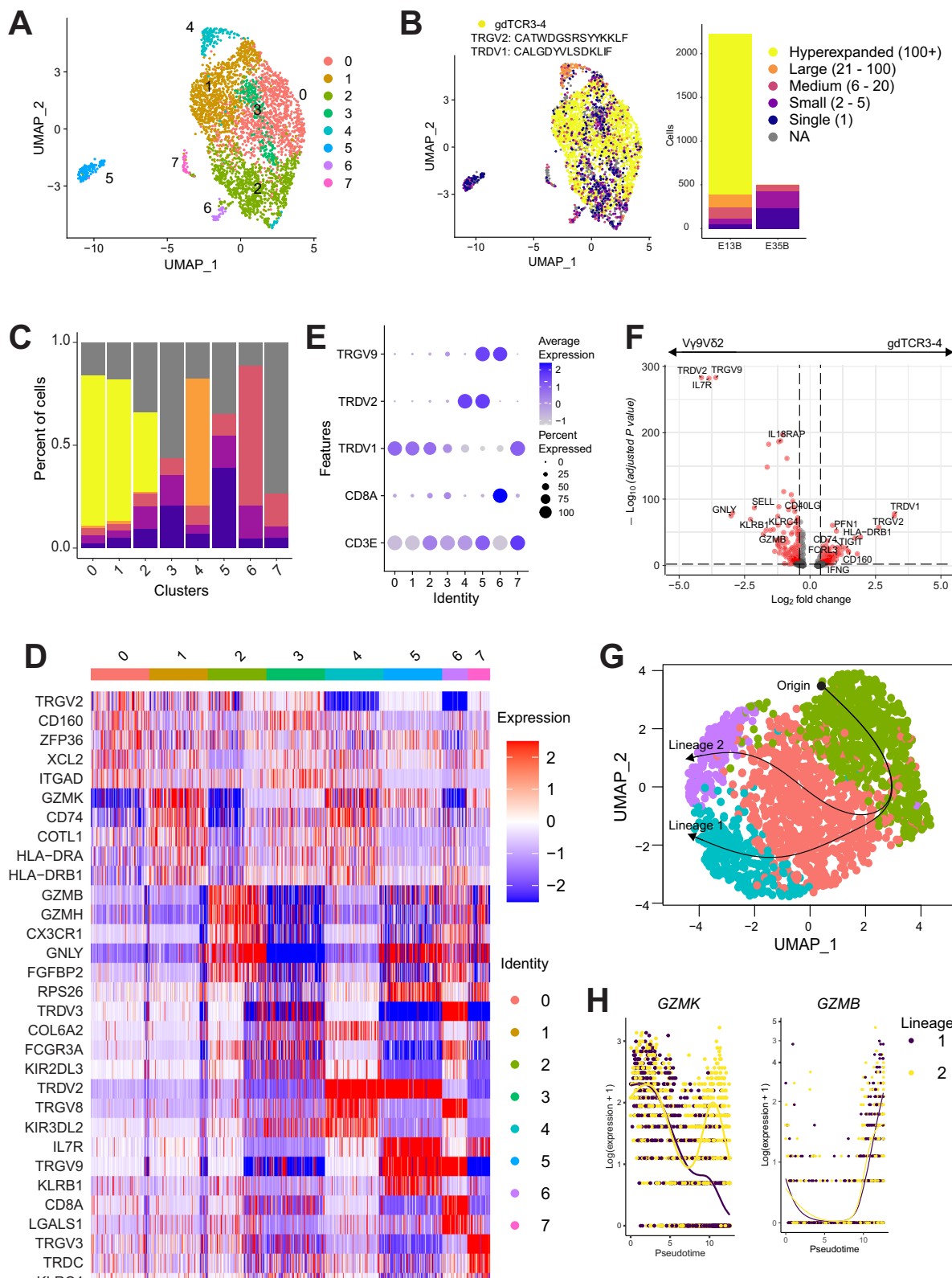

**Fig. 4 | Single cell analysis of clonally expanded γδ T cells after pembrolizumab treatment.** **A** UMAP of sorted γδ T cells from week six peripheral blood from patients E-013 and E-035. 2541 and 1105 cells are from E-013 and E-035 respectively. **B** Clonotypes were classified by the CDR3 nucleotide sequence and colored according to absolute number of cells. The γδTCR3-4 was the only hyperexpanded clonotype and highlighted as cells having CDR3 for *TRG*: CATWDGSRSYYKKLF, or *TRD*: CALGDYVLSDKLIF, or both. **C** Composition of each cluster by clonotype size.

**D** Heatmap of top five differentially expressed genes between each cluster. **E** Expression of select TRG and TRD chain usage within each cluster. **F** Volcano plot of differentially expressed genes between Vγ9Vδ2 cells found in cluster 5 and γδTCR3-4 cells. DEG were determined by two-sided Wilcoxon rank-sum test with log₂ FC > 0.4 and Bonferroni-adjusted *P* < 0.01. **G** γδTCR3-4 cells were subsetted and then reclustered with trajectory inference projection on top. **H** Smoother plots of *GZMK* and *GZMB* between two lineages over pseudotime is shown.

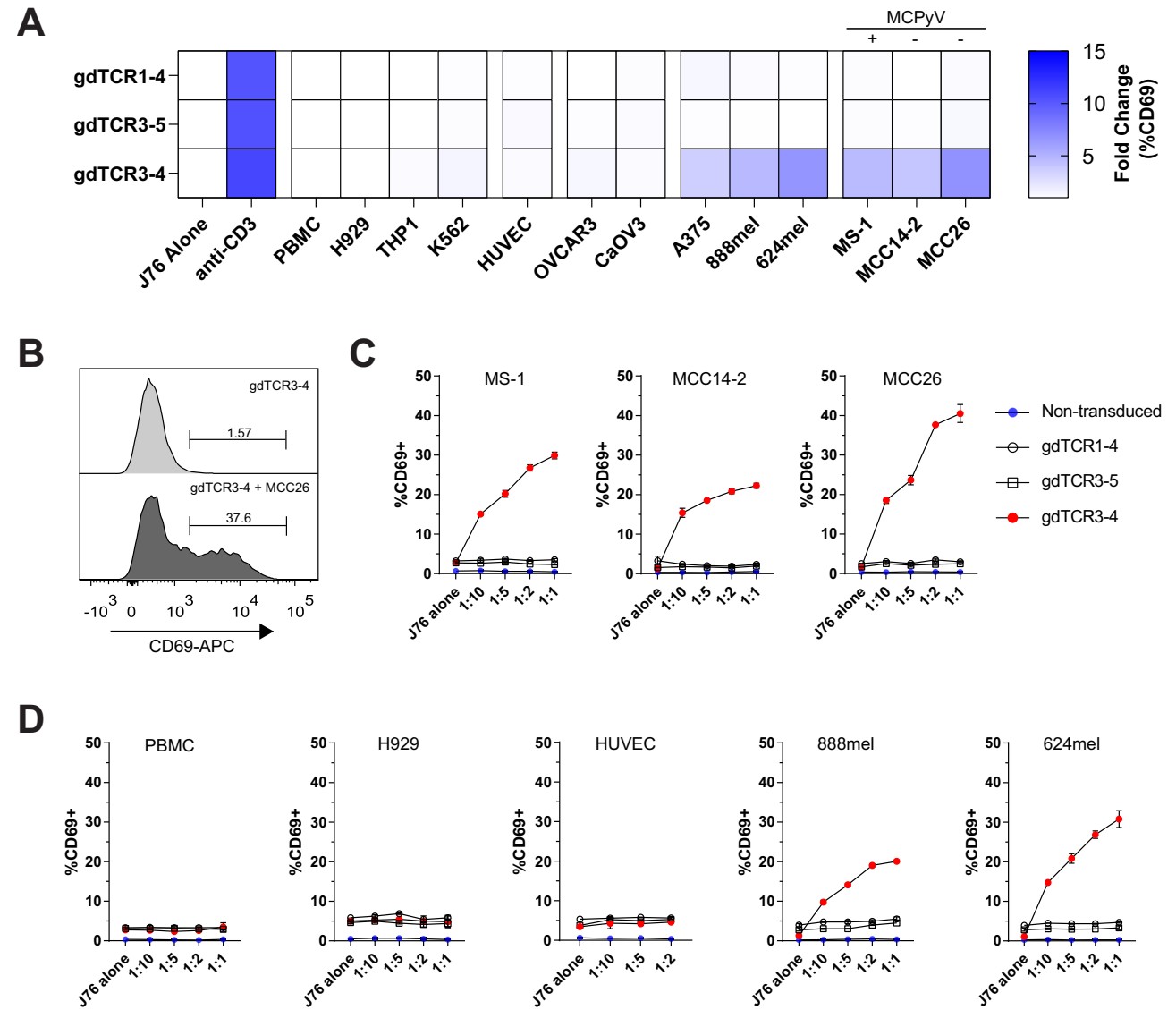

**Fig. 5 | γδTCR3-4 recognizes multiple Merkel cancer cell lines. A** The clonally expanded γδ TCR from patient E-013 was cloned and expressed into Jurkat76 cells (γδTCR3-4) along with mis-paired control cells (γδTCR1-4 and γδTCR3-5). Transduced γδTCR Jurkat76 cells were screened for reactivity against PBMCs and various cell lines including myeloma H929, leukemic cell lines THP1 and K562, human umbilical vein endothelial cells (HUVEC), ovarian cancer cell lines OVCAR3 and CaOV3, melanoma cell lines A375, 888mel and 624mel and Merkel cancer cell lines MS-1, MCC14-2 and MCC26. Fold change in CD69 expression was normalized by dividing the co-culture condition by the staining from Jurkat76 cells alone condition. MCC lines were denoted as negative (-) or positive (+) for Merkel cell polyomavirus (MCPyV). **B** Example histogram of CD69 expression on Jurkat76 γδTCR3-4 after co-culture with MCC26 at 1:2 target to effector ratio. **C** Jurkat76 cells were co-cultured with increasing numbers of MCC target cells. Non-transduced Jurkat76 cells are in blue, mis-paired γδ TCRs are in open circles and open squares, and γδTCR3-4 is in red circles. **D** Jurkat76 cells were co-cultured with increasing number of PBMCs, H929 myeloma, HUVEC and 888mel and 624mel melanoma cells. **C, D** Co-culture experiments were plated in duplicate and performed in two independent experiments. Data are presented as mean values +/- SD. Source data for (**A**, **C** and **D**) provided as a source data file.

(from 1.87 to 12.7%) in γδ T cell frequency within their tumor biopsies (Supplementary Fig. 4A, B). In addition, an increase in proliferating Ki-67[+] and TIM-3 expressing Vδ1 cells were detected in peripheral blood (Supplementary Fig. 4C, D), similar to the γδ T cell response from MCC patient E-013. Unfortunately, the TNBC patient did not respond to pembrolizumab treatment, which potentially could be attributed to a variety of reasons including immunosuppressive and regulatory cells within the tumor immune microenvironment. These findings, together with recent reports in bladder cancer[17] and MHC-class I-deficient cancers[18], suggest that γδ T cells may directly contribute to the anti-tumor response after PD-1 targeted immunotherapy in multiple disease settings.

## Discussion

In the tumor microenvironment, γδ T cells can have either pro- or anti-tumor functions[28]. In pre-clinical mouse models, γδ T cells have been shown to be a critical source of IFNγ during tumor surveillance[29]. Conversely, γδ T cell production of IL-17A and VEGF has been linked to angiogenesis, metastasis, and pro-tumor survival[30–32]. These dichotomous roles have also been reflected in human cancers with conflicting reports on the prognostic value of γδ T cells. In cancers such as breast, lung, and ovarian cancer, increased infiltration of γδ T cells is correlated with survival[33–35], while in other cancer types such as pancreatic and colorectal cancer, γδ T cell infiltration correlated with tumor development[36,37]. Our study demonstrates tumor-reactive Vγ2Vδ1 T cells expand after PD-1 blockade, express multiple markers

consistent with stimulation and provides a direct mechanism of how γδ T cells can improve patient survival.

A recent study by Gherardin et al. has shown that γδ T cells can have a beneficial prognostic impact on MCC patients. They found γδ T cells in MCC tumors with an exhausted phenotype as indicated by co-expression of PD-1 and LAG3. Similarly, we also found that tumor-infiltrating γδ T cells expressed high levels of co-inhibitory molecules, PD-1 and TIGIT. However, most of the MCC patients in the Gherardin et al. study were not treated with immunotherapy.

One interesting aspect of our study is the characterization of the γδ TCR repertoire pre- and post- pembrolizumab treatment. It has been previously shown that the adult Vδ1 TCR repertoire is highly focused on a few dominant clonotypes due to selection by viral infection and potentially cancer[38]. As a result of our γδ TCR sequencing, we captured a five-fold expansion of a γδ T cell clonotype in peripheral blood after six weeks of pembrolizumab treatment, which was followed by a 10-fold expansion in the frequency of γδ T cells in the tumor at week nine. We have also found approximately six-fold expansion of γδ T cells in the tumor of a TNBC patient. Previous reports have shown that clonal replacement of αβ T cells following PD-1 blockade originates from tumor-extrinsic sources, such as lymphoid organs or peripheral blood[39]. Similarly, we observe proliferation of clonally expanded Vδ1 T cells in the blood and expansion of γδ T cells in the tumor after pembrolizumab treatment. These results suggests that anti-PD-1 blockade is exerting its function on peripheral γδ T cells, which are subsequently recruited into the tumor, in a similar manner to αβ T cells.

Vδ1 T cells typically reside within epithelial tissue and are known to be involved in tumor surveillance[40]. Recognition of the inducible stress ligand MICA/B on malignant cells can be mediated by both NKG2D[41] as well as the Vδ1 TCR[5]. Additionally, Vδ1 T cells expanded from tumor infiltrates have been shown to have greater cytotoxic potential as compared to Vδ2 T cells[42]. The cytotoxicity of Vδ1 T cells is also more dependent on granzyme B and perforin-mediated mechanisms rather than antibody-dependent cellular cytotoxicity[43]. From our scRNA sequencing data, the Vδ1 T cells in peripheral blood expressed higher levels of multiple granzymes and perforin, which may also be indicative of cytotoxic potential and promotion of tumor clearance.

It has been previously demonstrated that upon CMV infection, there is upregulation of stress ligands such as EPCR[44] and ANXA2[45] that can be recognized by γδ T cells. While MCC is commonly associated with MCPyV infection, and all of the MCC patients within the INSPIRE cohort were virus-positive, it is clear that the ligand for γδTCR3-4 is not dependent on MCPyV infection for several reasons. First, of the three different MCC cell lines utilized in this study, only one cell line (MS-1) is MCPyV-positive[46,47]. Furthermore, the MCC26 cell line used for CRISPR experiments is MCPyV-negative, but still had the strongest reactivity to γδTCR3-4. Secondly, the three different melanoma cell lines A375, 624mel and 888mel are not associated with any viral infection but were still able to be recognized by γδTCR3-4. Lastly, the etiology of MCC from Australia is generally more associated with UV damage rather than McPyV infection[48]. This was reflected in the Gherardin et al. study, where two-thirds of the MCC patients were virus negative, and viral status was not correlated with γδ T cell infiltration.

Loss of antigen presentation can be a form of immune evasion and resistance mechanism to PD-1 blockade, for example, with mutations in *B2M*[49] or HLA loss of heterozygosity[50]. A recent report by de Vries et al. demonstrated that in patients with mismatch repair-deficient (MMR-d) cancers, those with *B2M* mutations had a higher number of Vδ1/3 T cells in the tumor compared to patients with wildtype *B2M*[18]. The authors were able to confirm that *B2M* mutated MMR-d cancers derived clinical benefit from treatment with anti-PD-1 therapy, which demonstrates that in tumors that lack MHC class I antigen presentation, other immune effectors such as γδ T cells can mediate response

to PD-1 blockade. Additionally, they showed that recognition of *B2M* mutated tumors by γδ T cells was mediated by NKG2D, and checkpoint blockade further increased the number of Vδ1/3 T cells in the tumor. However, in their study, checkpoint blockade also increased the amount of other tumor-infiltrating lymphocytes, including NK cells, CD8 and CD4 T cells. Therefore, they were unable to formally demonstrate whether the increase in γδ T cells was directly or indirectly influenced by checkpoint blockade. Because tumor recognition in our study was mediated by a defined γδ TCR, we were able to monitor clonal expansion in the blood and tumor post-treatment and exclude potential indirect methods of γδ T cell expansion such as cytokine signaling. However, both studies support that the antigen for immunosurveillance by γδ T cells is β2 microglobulin-independent.

To date, the applications of Vδ1 T cells as cancer immunotherapy have been focused on the ex vivo expansion of polyclonal Vδ1 T cells to be used as adoptive cell therapy[51]. These expanded Vδ1 T cells recognize tumor cells using multiple natural cytotoxicity receptors such as NKp30, NKp44 and NKp46, rather than through the TCR to mediate anti-tumor function[51]. As the Vδ1 T cells we describe in our study is a TCR clonotype that recognizes a Merkel cell cancer antigen, a potential development for cancer immunotherapy could be engineered TCR therapy in an HLA-unrestricted manner. Finally, our study provides the rationale to utilize γδ T cells for treating cancer variants that have lost expression of MHC class I and warrants further investigation of the interplay between γδ T cells and checkpoint blockade.

## Methods

### Clinical trial and patient details

The INvestigator-initiated Phase 2 Study of Pembrolizumab Immunological Response Evaluation (INSPIRE) is a single centre study approved by the Research Ethics Board at the Princess Margaret Cancer Centre and is registered at https://clinicaltrials.gov/ct2/show/NCT02644369. This trial was conducted in accordance with the Declaration of Helsinki and written informed consent was given by all patients. As previously reported, a total of 106 patients were recruited and enrolled in one of five cohorts: head and neck squamous cell carcinoma, triple-negative breast cancer, high-grade serous carcinoma, metastatic melanoma and mixed solid tumors[52–54]. All patients were naïve to anti-PD-1/PD-L1 therapy prior to enrollment, and were treated with 200 mg of pembrolizumab given every three weeks intravenously, for a maximum of two years. Details and characteristics of the MCC patients can be found in Supplementary Table 1.

### Tumor processing and dissociation

Tumor lesions were biopsied at baseline (within 28 days prior to study treatment) and on-treatment (week six or week nine of pembrolizumab treatment). Pooled core biopsies or tissue samples were minced into 2–4 mm³ fragments and then mechanically and enzymatically digested with the gentle MACS dissociator (Miltenyi Biotec, Catalog #130–093-235) and the human tumor dissociation kit (Miltenyi, Catalog #130–095-929). Freshly digested single cell suspensions were then used for flow cytometry analysis.

### Blood collection

Peripheral blood samples were collected in sodium heparin tubes at baseline, week 3, week 6, week 9, week 15 and every nine weeks thereafter and at the end-of-treatment.

### Flow cytometry and cell sorting

Cells were incubated with Fc receptor inhibitor (Thermo Fisher Scientific, Catalog #14-9161-73) and eFluor506 fixable viability dye (Thermo Fisher Scientific, Catalog #65-0866-14), followed by primary antibody cocktails, and then fixed with 4% paraformaldehyde. Primary antibody stain and fixation was performed for 30 min at 4 °C and protected from light. For biotinylated antibodies, a 15 min incubation

with streptavidin-conjugated fluorochrome was added after antibody staining. For intracellular staining, cells were fixed, permeabilized and stained with the FoxP3 Transcription Factor set (Thermo Fisher Scientific, Catalog #00-5523-00). Flow cytometry data was then acquired on a LSR Fortessa and analyzed using FlowJo v10 (BD Biosciences). For detailed information on optimized flow cytometry panels, refer to Table 2.

For fluorescent-activated cell sorting (FACS), cells were incubated with Fc receptor inhibitor and eFluor506 fixable viability dye, and then stained with anti-CD3-PE-Cy7, anti-γδ TCR-PE, anti-CD8-FITC, anti-CD4-APC, and sorted with FACS Aria Fusion (BD Biosciences).

### Cell lines

The following MCC lines were purchased from Sigma-Aldrich as part of the European Collection of Authenticated Cell Cultures: MCC14/2 (Catalog #10092303), MCC26 (Catalog #10092304), MS-1 (Catalog #09111802). The MCC lines were cultured in RPMI-1640 (Gibco) and supplemented with 20% fetal calf serum (FCS), L-glutamine, and HEPES. The cell line CaOV3 (Catalog #HTB-75) and Human Umbilical Vein Endothelial Cells (Catalog #CRL1730) were purchased from the American Type Culture Collection. HUVEC cells were cultured in Human Large Vessel Endothelial Cell Basal Medium (Thermo Fisher Scientific, Catalog #M200500) and supplemented with Large Vessel Endothelial Supplement (Thermo Fisher Scientific, Catalog #A1460801). The cell lines OVCAR3, H929 and A375 were gifts from Dr. T. Mak (Princess Margaret Cancer Centre, Toronto, Canada). The CaOV3 and A375 cell lines were cultured in DMEM and supplemented with 10% FCS, and L-glutamine. OVCAR3 cells were cultured in RPMI-1640 and supplemented with 10% FCS, 0.01 mg/ml bovine insulin, and L-glutamine. Jurkat76 cells (a gift from Dr. M. Heemskerk, Leiden University Medical Center, Leiden, the Netherlands) were cultured in RPMI-1640 and supplemented with 10% FCS, and L-glutamine. The melanoma cell lines 624mel and 888mel (gifts from Dr. S. Rosenberg, National Cancer Institute) and H929 cells were maintained in RPMI-1640, and supplemented with 10% FCS and L-glutamine. All cell lines were maintained in presence of penicillin, streptomycin, and gentamicin. Cell lines received as gifts were not authenticated.

### Bulk TCR sequencing

Genomic DNA from sorted γδ T cells was extracted with AllPrep DNA/RNA (Qiagen, Catalog #80284). For FFPE tissue, DNA was extracted using AllPrep DNA/RNA FFPE (Qiagen, Catalog # 80234). DNA was sent for *TRG* or *TRA/D* sequencing by the Adaptive Biotechnologies immunoSEQ assay where they performed high throughput sequencing of the TCR variable regions after multiplexed PCR amplifications that targeted all potential VDJ recombinations. Synthetic TCR templates were spiked in to computationally control for PCR amplification bias. Reads were aligned and filtered for in-frame CDR3 sequences. Downstream analysis was performed with *immunarch* package (v0.6.9).

### Single cell RNA and TCR sequencing

γδ T cells were sorted and then single cell libraries were created with 10x Genomics Chromium 5′ Kit v2 by the Princess Margaret Genomics Centre. γδ TCR libraries were prepared as previously described[55]. For data pre-processing, low quality cells with less than 300 unique feature counts or more than 8% mitochondrial counts and cell doublets with greater than 4000 unique feature counts were removed. Analysis of single cell data was performed using the R packages *Seurat*[56] (v4.1.1) and *scRepertoire*[57,58] (v1.6.0), *SingleCellExperiment* (v1.18.0), *dittoseq* (v1.9.1), *slingshot* (v.2.4.0) and *EnhancedVolcano* (v1.14.0).

### Jurkat76 transduction of γδ TCR

Jurkat76 cells were transduced with individual TCRγ and TCRδ chains as previously reported[59]. Briefly, full length TCRγ (TRG) or

TCRδ (TRD) chains were synthesized (Invitrogen GeneArt Gene Synthesis) from partial CDR3 variable region reads and cloned into pMX expression vector for retroviral packaging. Individual TRG and TRD genes were transduced into Jurkat76 using spinfection and after 5 days, TCR transfectants were selected to purity (>95% purity) using CD3 Microbeads (Miltenyi Biotec, Catalog #130-050-101).

### Co-culture of Jurkat76.gdTCR and cancer cell lines

Tumor cells were trypsinized, washed and then 25,000 cells were seeded on a 96 well flat bottom plate and incubated overnight. Positive control wells were pre-coated with 5 μg/mL of anti-CD3 OKT3 (Biolegend, Catalog #317326) overnight at 37 °C and then washed. Afterwards, 50,000 transduced Jurkat76 cells or non-transduced control cells were added. Following a 16–20-h incubation, cells were stained with viability dye, anti-CD3-PE (BD Biosciences, Catalog # 566683) and anti-CD69-APC (Biolegend, Catalog # 310910). For calculating fold change of %CD69, Jurkat76 cells were gated by CD3 expression and then frequency of CD69 was normalized by $\frac{\%CD69\,of\,Condition}{\%CD69\,of\,Jurkat76alone}$. For dose-dependent TCR recognition of cancer cell lines, 5000, 10,000, 25,000 and 50,000 cancer cells were co-cultured with 50,000 Jurkat76 cells.

### CRISPR-Cas9 knockout of Merkel cancer cell line

Chemically modified synthetic guide RNAs and *S. pyogenes* Cas9 nuclease were purchased from Synthego. Target sequences can be found in Supplementary Table 6. To create Cas9-ribonucleoproteins (RNPs), 180 pmol of multi-guide sgRNAs were combined with 20 pmol of Cas9 and incubated at room temperature for 10 min. RNPs were electroporated into MCC26 cells using the Nucleofector 2b and Amaxa Cell Line Nucleofector Kit V (Lonza, Catalog #VCA-1003) with the D-023 program. After electroporated cells recovered and were passaged once, knockout cells were purified by FACS sorting.

### Statistics and reproducibility

Statistical analysis was performed in GraphPad Prism 8 and within R version 4.2.0. Statistical tests between two groups were performed using by two-sided Wilcoxon rank-sum and multiple testing is adjusted with Bonferroni correction. $p < 0.01$ was considered statistically significant. No statistical method was used to predetermine sample size. On-treatment biopsy sample from patient E-020 was unavailable due to insufficient tumor material. Patient E-033 progressed and went off trial prior to collection of on-treatment tumor biopsy and week 6 blood sample. The experiments were not randomized. The investigators were not blinded to allocation during experiments and outcome assessment.

### Reporting summary

Further information on research design is available in the Nature Portfolio Reporting Summary linked to this article.

## Data availability

The single cell gene expression count matrix and V(D)J calls generated in this study are available in the Supplementary Data file. The raw scRNA and TCR sequencing data can be found in the European Genome-phenome Archive under accession code EGAD50000000137 (https://ega-archive.org/datasets/EGAD50000000137). The sequencing data is available under controlled access in compliance with patient consent for data sharing and access can be obtained by approval from the University Health Network Data Access Committee (email: dac@uhn.ca). The bulk TCR γ and δ sequencing data generated in this study are available in the immuneACCESS database (https://doi.org/10.21417/SCL2023NC). Source data are provided as a Source Data file in this paper. Source data are provided with this paper.

## Code availability

The code for scRNAseq and TCR analysis is available on Zenodo (https://zenodo.org/records/10119047).

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

## Acknowledgements

We would like to thank the patients and their families for their participation in the INSPIRE clinical trial. This study was supported by the Terry Fox Research Foundation (Terry Fox Translational Research Program iTNT #1060) and Terry Fox Research Institute (New Frontiers Program TFRI # 1064). Merck Canada Inc., Kirkland, QC, Canada, kindly provided pembrolizumab. We thank the PM Flow Cytometry Facility and PM Genomics Centre at the Princess Margaret Cancer Centre for their assistance. Lastly, we would like to thank Stephanie Wong, Charlotte Lo and Meghan Kates for critical reading and editing of this manuscript.

## Author contributions

S.C.L. and P.S.O. designed and implemented the study. T.J.P. and P.S.O. secured funding. P.L.B., A.S. and A.R.A.R. accrued patients. B.X.W., D.L.C. and S.B.H. coordinated sample collection. S.C.L., D.L., R.G., B.N. and C.R.G. performed experiments. D.L., M.S.P. and N.H. provided technical expertise. S.Y.C.Y. and T.J.P. provided expertise for bioinformatic analyses. S.C.L. and P.S.O. wrote the manuscript, which all authors reviewed and edited.

## Competing interests

D.L.C. is currently an employee of AstraZeneca and had been employed by Ipsen and Pfizer after his involvement in this project. T.J.P. provides consultation for Illumina, Merck, Chrysalis Biomedical Advisors and the Canadian Pension Plan Investment Board (compensated); and receives research support (institutional) from Roche/Genentech. A.S. provides consultation for Merck (compensated), Bristol-Myers Squibb (compensated), Novartis (compensated), and Oncorus (compensated); and receives research support (institutional) from Novartis, Bristol-Myers Squibb, Symphogen AstraZeneca/Medimmune, Merck, Bayer, Surface Oncology, Northern Biologics, Janssen Oncology/Johnson & Johnson, Array Biopharma. N.H. has received research funding from Takara Bio and served as a consultant for Takara Bio. A.R.A.R. provides consultation for Lilly (compensated), Merck (compensated), and Boehringer-Ingelheim (compensated); receives honoraria from Boehringer-Ingelheim; and receives research support (institutional) from CASI Pharmaceuticals, Boehringer-Ingelheim, Lilly, Novartis, Deciphera, Karyopharm, Pfizer, Roche/Genentech, Boston Biomedical, Bristol-Myers Squibb, AstraZeneca/MedImmune, Amgen, GlaxoSmithKline, Blueprint Medicines, Merck, Abbvie, and Adaptimmune. P.S.O. is an SAB member for Symphogen Inc and Providence Therapeutics. P.S.O, A.R.A.R., N.H., D.L. and S.C.L. have filed a provisional patent related to this work (US63/459,683). The remaining authors declare no competing interests.
