## [Peer Review File · Nature Communications]

Tumor specific reactive $\gamma\delta$ T cells expand contribute to a complete response and respond to PD-1 blockade in a Merkel cell carcinoma patientEditorial Note: This manuscript has been previously reviewed at another journal that is not operating a transparent peer review scheme. This document only contains reviewer comments and rebuttal letters for versions considered at *Nature Communications*.

REVIEWERS' COMMENTS

Reviewer #1 (Remarks to the Author):

The authors acknowledged the limitations of their study (as highlighted in the review process in Nature) and provided a new supplementary figure to clarify one of the outstanding issues. The other main issue appears to be technically very difficult to address. Given the clinical relevance of the main findings of the study, I consider it suitable for publication in Nat Comms.

Reviewer #2 (Remarks to the Author):

I think the final sentence of the abstract should be: "Our results demonstrate that innate-like T cells can also contribute to a tumor specific response after PD-1 blockade therapy." I'm also worried about the title given that this observation is limited to a fraction of patients analyzed. I think if "can" or some other way to convey the fact this response to PD-1 blockade does not always happen would help to make the title more accurate.

Otherwise, my comments have been adequately addressed. I appreciate the additional clinical information, the efforts made when it comes to the specificity of the gd TCR, and the inclusion of additional anti-PD1 treated samples. It's interesting that there is also reactivity with melanoma cell lines.

Referee #1 (Remarks to the Author):

The authors acknowledged the limitations of their study (as highlighted in the review process in Nature) and provided a new supplementary figure to clarify one of the outstanding issues. The other main issue appears to be technically very difficult to address. Given the clinical relevance of the main findings of the study, I consider it suitable for publication in Nat Comms.

We thank the referee for their time and review of our manuscript.

Referee #2 (Remarks to the Author):

I think the final sentence of the abstract should be: “Our results demonstrate that innate-like T cells can also contribute to a tumor specific response after PD-1 blockade therapy.” I’m also worried about the title given that this observation is limited to a fraction of patients analyzed. I think if “can” or some other way to convey the fact this response to PD-1 blockade does not always happen would help to make the title more accurate.

Otherwise, my comments have been adequately addressed. I appreciate the additional clinical information, the efforts made when it comes to the specificity of the gd TCR, and the inclusion of additional anti-PD1 treated samples. It’s interesting that there is also reactivity with melanoma cell lines.

We thank the reviewer for their time and are glad they found our work interesting. We have amended the final sentence of the abstract to:

“Our results demonstrate that innate-like T cells could also contribute to an anti-tumor response after PD-1 blockade.”

We have also changed the title to:

“Tumor reactive $\gamma\delta$ T cells contribute to a complete response to PD-1 blockade in a Merkel cell carcinoma patient”.

We hope these changes can more accurately describe the observations made in the patient analyzed.